# PathologyGAN: Learning deep representations of cancer tissue

**Adalberto Claudio Quiros** [1]                A.CLAUDIO-QUIROS.1@RESEARCH.GLA.AC.UK
[1] *University of Glasgow, School of Computing Science*
**Roderick Murray-Smith** [1]                RODERICK.MURRAY-SMITH@GLASGOW.AC.UK
[1] *University of Glasgow, School of Computing Science*
**Ke Yuan** [1]                KE.YUAN@GLASGOW.AC.UK
[1] *University of Glasgow, School of Computing Science*

## Abstract

We apply Generative Adversarial Networks (GANs) to the domain of digital pathology. Current machine learning research for digital pathology focuses on diagnosis, but we suggest a different approach and advocate that generative models could drive forward the understanding of morphological characteristics of cancer tissue. In this paper, we develop a framework which allows GANs to capture key tissue features and uses these characteristics to give structure to its latent space. To this end, we trained our model on 249K H&E breast cancer tissue images, extracted from 576 TMA images of patients from the Netherlands Cancer Institute (NKI) and Vancouver General Hospital (VGH) cohorts.

We show that our model generates high quality images, with a Fréchet Inception Distance (FID) of 16.65. We further assess the quality of the images with cancer tissue characteristics (e.g. count of cancer, lymphocytes, or stromal cells), using quantitative information to calculate the FID and showing consistent performance of 9.86. Additionally, the latent space of our model shows an interpretable structure and allows semantic vector operations that translate into tissue feature transformations. Furthermore, ratings from two expert pathologists found no significant difference between our generated tissue images from real ones. The code, generated images, and pretrained model are available at https://github.com/AdalbertoCq/Pathology-GAN

**Keywords:** Generative Adversarial Networks, Digital Pathology.

## 1. Introduction

Cancer is a disease with extensive heterogeneity, where malignant cells interact with immune cells, stromal cells, surrounding tissues and blood vessels. Histological images, such as haematoxylin and eosin (H&E) stained tissue microarrays (TMAs), are a high-throughput imaging technology used to study such diversity. Despite its growing popularity, computational analysis of TMAs often lacks analysis of other omics data of the same cohort. Consequently, cellular behaviours and the tumour microenvironment recorded in TMAs remain poorly understood.

The motivation for our work is to develop methods that could lead to a better understanding of phenotype diversity between/within tumors. We hypothesize that this diversity could be quite substantial given the highly diverse genomic and transcriptomic landscapes

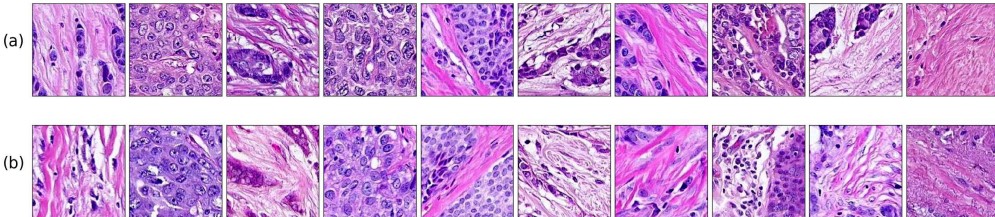

Figure 1: (a): Images ($224 \times 224$) from PathologyGAN trained on H&E breast cancer tissue. (b): Real images, Inception-V1 closest neighbor to the generated above.

observed in large scale molecular profiling of tumors across multiple cancer types (Ciriello et al., 2013). We argue that representation learning with GAN-based models is the most promising to achieve our goal for the two following reasons:

1. By being able to generate high fidelity images, a GAN could learn the most relevant descriptions of tissue phenotype.

2. The continuous latent representation learned by GANs could help us quantify differences in tissue architectures free from supervised information.

In this paper, we propose to use Generative Adversarial Networks (GANs) to learn representations of entire tissue architectures and define an interpretable latent space (e.g. colour, texture, spatial features of cancer and normal cells, and their interaction). To this end, we present the following contributions:

1. We propose PathologyGANs to generate high fidelity cancer tissue images from an structured latent space. The model combines BigGAN (Brock et al., 2018), StyleGAN (Karras et al., 2018) and Relativistic Average Discriminator (Jolicoeur-Martineau, 2018).

2. We assess the quality of the generated images through two different methods: convolutional Inception-V1 features and prognostic features of the cancer tissue (such as counts and densities of different cell types (Beck et al., 2011; Yuan et al., 2012). Both features are benchmarked with the Fréchet Inception Distance (FID). The results show that the model captures pathologically meaningful representations, and when evaluated by expert pathologists, generated tissue images are not distinct from real tissue images.

3. We show that our model induces an ordered latent space based on tissue characteristics, this allows to perform linear vector operations that transfer into high level tissue image changes.

## 2. Background

GANs (Goodfellow et al., 2014) are generative models that are able to learn high fidelity and diverse data representations from a target distribution. This is done with a generator, $G(z)$,

that maps random noise, $\boldsymbol{z} \sim p_{\boldsymbol{z}}(z)$, to samples that resemble the target data, $\boldsymbol{x} \sim p_{\text{data}}(\boldsymbol{x})$, and a discriminator, $D(x)$, whose goal is to distinguish between real and generated samples. The goal of a GAN is find the equilibrium in the min-max problem:

$$\min_G \max_D V(D, G) = \mathbb{E}_{\boldsymbol{x} \sim p_{\text{data}}(\boldsymbol{x})}[\log D(\boldsymbol{x})] + \mathbb{E}_{\boldsymbol{z} \sim p_{\boldsymbol{z}}(\boldsymbol{z})}[\log(1 - D(G(\boldsymbol{z})))]. \tag{1}$$

Since its introduction, modeling distributions of images has become the mainstream application for GANs, firstly proposed by (Radford et al., 2016). State-of-the-art GANs such as BigGAN (Brock et al., 2018) and StyleGAN (Karras et al., 2018) have recently shown impressive high-resolution images, and proposed solutions like Spectral Normalization GANs (Miyato et al., 2018), Self-Attention GANs (Zhang et al., 2018), and also BigGAN achieved high diversity images in data sets like ImageNet (Deng et al., 2009) with 14 million images and 20 thousand different classes.

At the same time, evaluating these models has been a challenging task. Many different metrics such as Inception Score (IS) (Salimans et al., 2016), Fréchet Inception Distance (FID) (Heusel et al., 2017), Maximum Mean Discrepancy (MMD) (Gretton et al., 2012), Kernel Inception Distance (KID) (Binkowski et al., 2018), and 1-Nearest Neighbor classifier (1-NN) (Lopez-Paz and Oquab, 2017) have been proposed to do so, and thorough empirical studies (Xu et al., 2018; Barratt and Sharma, 2018) have shed some light on the advantages and disadvantages of each them. However, the selection of a feature space is crucial for using these metrics.

Currently, machine learning approaches to digital pathology have been focusing on building classifiers to achieve pathologist-level diagnosis (Esteva et al., 2017; Wei et al., 2019; Han et al., 2017), and assisting in the decision process through computer-human interaction (Cai et al., 2019).

Recently, there has been an increasing interest in applying GANs to solve a range of specific tasks in digital pathology, including staining normalization (Ghazvinian Zanjani et al., 2018), staining transformation (Rana et al., 2018; Xu et al., 2019), and nuclei segmentation (Mahmood et al., 2018). Together with deep learning-based classification frameworks (Esteva et al., 2017; Ardila et al., 2019), these advances offer hope for better disease diagnosis than standard pathology (Niazi et al., 2019). Deep learning approaches have a lack of interpretability, which is a major limiting factor in making a real impact in clinical practice.

For breast cancer, traditional computer vision approaches such as (Beck et al., 2011) and (Yuan et al., 2012) have identified correlation between morphological features of cells and patient survival. Based these findings, we propose PathologyGAN as an approach to learn clinically/pathologically meaningful representations within the cancer tissue images.

## 3. PathologyGAN

We use BigGAN from (Brock et al., 2018) as a baseline architecture and introduced changes which empirically improved the Fréchet Inception Distance (FID) and the structure of the latent space. We followed the same architecture as BigGAN, employed Spectral Normalization in both generator and discriminator, self attention layers, and we also use orthogonal initialization and regularization as mentioned in the original paper.

We make use of the Relativistic Average Discriminator (Jolicoeur-Martineau, 2018), where the discriminator's goal is to estimate the probability of the real data being more

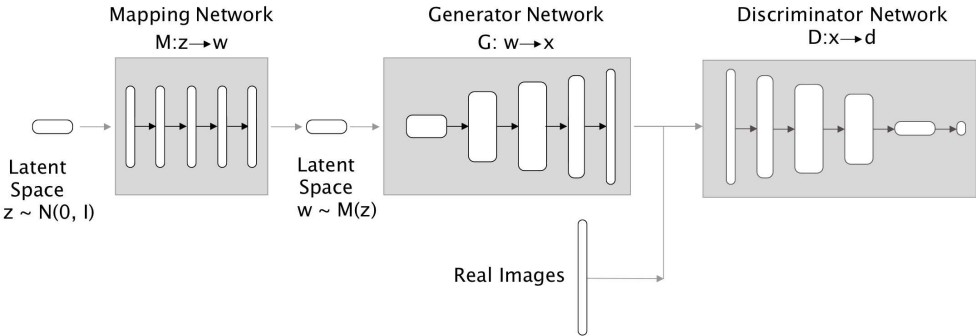

Figure 2: High level architecture of PathologyGAN. We include details of each modules architecture in the Appendix I

realistic than the fake. We take this approach instead of following the Hinge loss (Lim and Ye, 2017) as the GAN objective. We find that this change makes the model convergence faster and produce higher quality images. Images using the Hinge loss did not capture the morphological structure of the tissue (we provide examples of these results in the Appendix B). The discriminator, and generator loss function are formulated as in Equations 2 and 3, where $\mathbb{P}$ is the distribution of real data, $\mathbb{Q}$ is the distribution for the fake data, and $C(x)$ is the non-transformed discriminator output or critic:

$$L_{Dis} = -\mathbb{E}_{x_r \sim \mathbb{P}} \left[ \log \left( \tilde{D}(x_r) \right) \right] - \mathbb{E}_{x_f \sim \mathbb{Q}} \left[ \log \left( 1 - \tilde{D}(x_f) \right) \right], \tag{2}$$

$$L_{Gen} = -\mathbb{E}_{x_f \sim \mathbb{Q}} \left[ \log \left( \tilde{D}(x_f) \right) \right] - \mathbb{E}_{x_r \sim \mathbb{P}} \left[ \log \left( 1 - \tilde{D}(x_r) \right) \right], \tag{3}$$

$$\tilde{D}(x_r) = \text{sigmoid} \left( C(x_r) - \mathbb{E}_{x_f \sim \mathbb{Q}} C(x_f) \right), \tag{4}$$

$$\tilde{D}(x_f) = \text{sigmoid} \left( C(x_f) - \mathbb{E}_{x_r \sim \mathbb{P}} C(x_r) \right). \tag{5}$$

Additionally, we introduce two elements from StyleGAN (Karras et al., 2018) with the purpose of allowing the generator to freely optimize the latent space and find high-level features of the cancer tissue. First, a mapping network $M$ composed by four dense ResNet layers (He et al., 2015), placed after the latent vector $z \sim \mathcal{N}(0, I)$, with the purpose of allowing the generator to find the latent space $w \sim M(z)$ that better disentangles the latent factors of variation. Secondly, style mixing regularization, where two different latent vectors $z1$ and $z2$ are run into the mapping network and fed at the same time to the generator, randomly choosing a layer in the generator and providing $w1$ and $w2$ to the different halves of the generator (e.g. on a generator of ten layers and being six the randomly selected layer, $w1$ would feed layers one to six and $w2$ seven to ten). Style mixing regularization encourages the generator to localize the high level features of the images in the latent space. We also use adaptive instance normalization (AdaIN) on our models, providing the entire latent vectors.

We use the Adam optimizer(Kingma and Ba, 2015) with $\beta_1 = 0.5$ and same learning rates of 0.0001 for both generator and discriminator, the discriminator takes 5 steps for

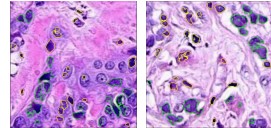

Figure 3: CRImage identifies different cell types in our generated images. Cancer cells are highlighted with a green color, while lymphocytes and stromal cells are highlighted in yellow.

each of the generator. Each model was trained on an NVIDIA Titan RTX 24 GB for approximately 72 hours.

To train our model, we utilize two H&E breast cancer databases from the Netherlands Cancer Institute (NKI) cohort and the Vancouver General Hospital (VGH) cohort with 248 and 328 patients respectively (Beck et al., 2011). Each of them include TMA images, along with clinical patient data such as survival time, and estrogen-receptor (ER) status. The original TMA images all have a resolution of $1128 \times 720$ pixels, and we split each of the images into smaller patches of $224 \times 224$, and allow them to overlap by 50%. We also perform data augmentation on these images, a rotation of 90°, and 180°, and vertical and horizontal inversion. We filter out images in which the tissue covers less than 70% of the area. In total this yields a training set of 249K images and a test set of 62K.

## 4. Results

### 4.1. Image quality analysis

We evaluate our models using the Fréchet Inception Distance (FID), we take the usual procedure of using convolutional features of a Inception-V1 network. As additional assessment, we use cellular information of the tissue image to calculate the FID score, the motivation behind this approach is to ensure that our models capture meaningful and faithful representations of the tissue.

The CRImage tool (Yuan et al., 2012) uses an SVM classifier to provide quantitative information about tumor cellular characteristics in cancer tissue. This approach allows us to gather pathological information in the images, namely the number of cancer cells, the number of other types of cells (such as stromal or lymphocytes), and the ratio of tumorous cells per area. We use this information as features to calculate the FID metric. Figure 3

| Model | Inception FID | CRImage FID |
|-------|---------------|-------------|
| PathologyGAN | 16.65±2.5 | 9.86±0.4 |

Table 1: Evaluation of PathologyGANs. Mean and standard deviations are computed over three different random initializations. The low FID scores in both feature space suggest consistent and accurate representations.

displays an example of how the tool captures the different cells in the generated images, such as cancer cells, stromal cells, and lymphocytes.

We evaluate our model with the FID score, generating 5000 fake images, and randomly selecting 5000 real images. We also use both approaches for feature space selection, using CRImage cell classifier and the convolutional features of an Inception-V1.

Table 1 shows that our model is able to achieve a accurate characterization of the cancer tissue. Using the Inception feature space, FID shows a stable representation for all models with values similar to ImageNet models of BigGAN (Brock et al., 2018) and SAGAN (Zhang et al., 2018)), with FIDs of 7.4 and 18.65 respectively or StyleGAN (Karras et al., 2018) trained on FFHQ with FID of 4.40. Using the CRImage cellular information as feature space, FID shows again close representations to real tissue.

## 4.2. Analysis of latent representations

In this section we focus on the PathologyGAN's latent space, exploring the impact of introducing a mapping network in the generator and using style mixing regularization. We include a complete comparison on using these two features in the Appendix C. Here we will provide examples of its impact on linear interpolations and vector operations on the latent space $z$, as well as visualizations on the latent space $w$.

In Figure 4 we capture how the latent space $w$ has a structure that shows direct relationship with the number of cancer cells in the tissue. We generated $10,000$ images and use CRImage to extract the count of cancer cells in the tissue, using this information we created 8 different classes that account for counts of cancer cells in the tissue image, and consecutively we label each image with the corresponding class. Along with each tissue image we have the corresponding latent vector $w$, we used UMAP (McInnes et al., 2018) to project it to two dimensional space.

In each of the sub-figures (a-h) in Figure 4, we provide a density plot of latent vectors $w$ for each class, (a) corresponds to generated images with the lowest count of cancer cells in the tissue and (h) to images with the largest, as we increase the number of cancer cells in the image, the density of the latent vectors $w$ moves from quadrant $IV$ to quadrant $II$ in the UMAP representation.

We also found that linear interpolations between two latent vectors $z$ have better feature transformations when the mapping network and style mixing regularization are introduced. Figure 5 shows linear interpolations in latent space $z$ between images with malignant tissue and benign tissue. (a) corresponds to a model with a mapping network and style mixing regularization and (b) to a model without those features, we can see that transitions on (a) include an increasing population of cancer cells rather than the fading effect observed in images of (b). This result indicates that (a) better translates interpolations in the latent space, as real cells do not fade away.

Finally, we performed linear vector operations in $z$, that translated into semantic image features transformations. In Figure 6 we provide examples of three vector operations that result into feature alterations in the images. This evidence shows further support on the relation between a structured latent space and tissue characteristics.

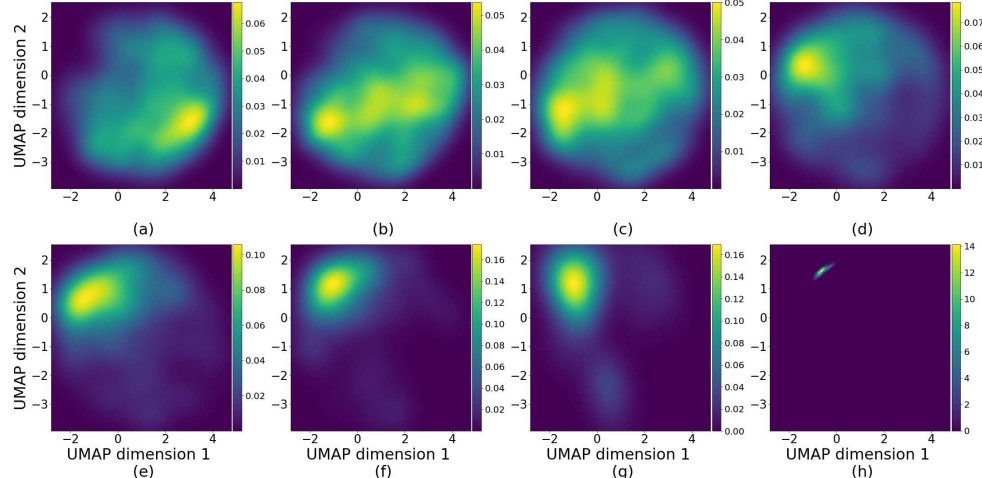

Figure 4: Density plots of samples on the UMAP reduced representation of the latent space $w$. Each subfigure (a-h) belongs to samples of only one class, where each class represents a range of counts of cancer cells in the tissue image. (a) accounts for images with the lowest number of cancer cells and (h) corresponds to images with the largest count, subfigures from (a) to (h) belong to increasing number of cancer cells. This figure shows how increasing the number of cancer cells in the tissue image corresponds to moving the latent vectors from regions of quadrant $IV$ to quadrant $II$.

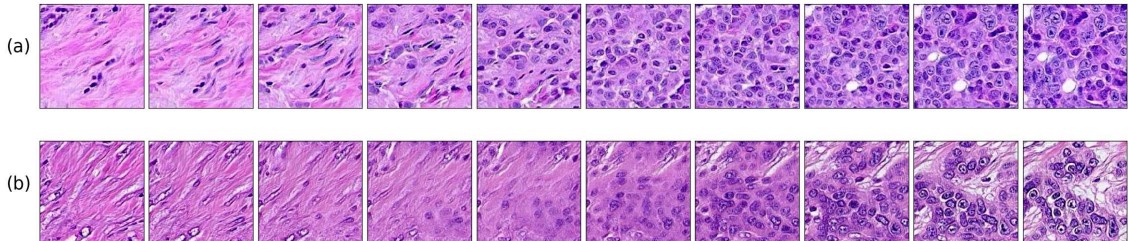

Figure 5: Linear interpolation in the latent space $z$ from a benign (less cancer cells, left end) to a malignant tissue (more cancer cells, right end). (a) PathologyGAN model interpolations with a mapping network and style mixing regularization. (b) PathologyGAN model interpolations without a mapping network and style mixing regularization. (a) includes an increasing population of cancer cells rather than a fading effect from model (b), this shows that model (a) better translates high level features of the images from the latent space vectors.

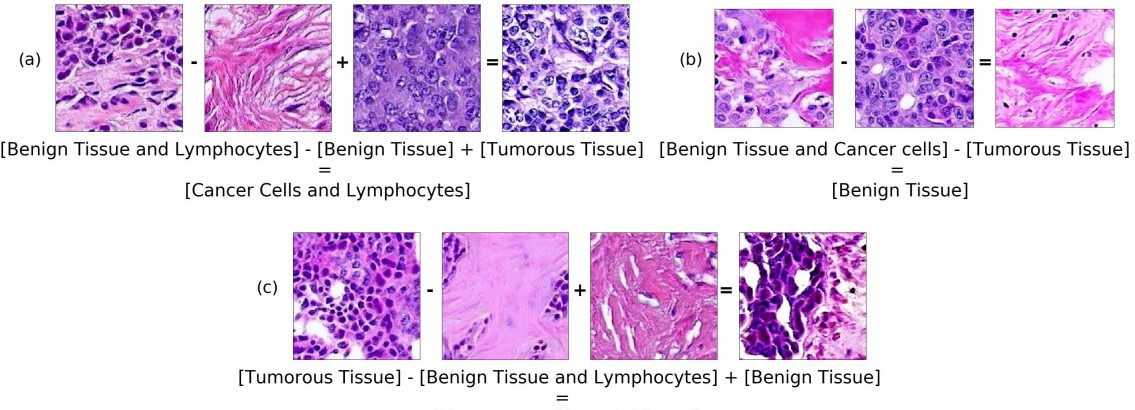

[Benign Tissue and Lymphocytes] - [Benign Tissue] + [Tumorous Tissue]  [Benign Tissue and Cancer cells] - [Tumorous Tissue]
=
[Cancer Cells and Lymphocytes]                                                          [Benign Tissue]

[Tumorous Tissue] - [Benign Tissue and Lymphocytes] + [Benign Tissue]
=
[Tumorous or Necrotic Tissue]

Figure 6: Linear vector operations on the latent space $z$ translate into image feature transformations. We gather latent vectors that generate images with different high level features and perform linear operations on the vectors before we feed the generator, resulting into semantic translations of the characteristics of the images. We perform the following operations (a) Benign tissue and lymphocytes - benign tissue + tumorous tissue = cancer cells and lymphocytes, (b) Benign tissue with patches of cancer cells - tumorous = benign tissue, and (c) Tumorous tissue with lymphocytes - benign tissue with lymphocytes + benign tissue = tumorous or necrotic tissue.

### 4.3. Pathologists' results

To demonstrate that the generated images can sustain the scrutiny of clinical examination, we asked two expert pathologists to take two different tests, setup as follows:

- Test I: 25 Sets of 8 images - Pathologists were asked to find the only fake image in each set.

- Test II: 50 Individual images - Pathologists were asked to rate all individual images from 1 to 5, where 5 meant the image appeared the most real.

In total, each of the pathologists classified 50 individual images and 25 sets of 8 images. We chose fake images in two ways, with half of them hand-selected and the other half with fake images that had the smallest Euclidean distance to real images in the convolutional feature space (Inception-V1). All the real images are randomly selected between the three closest neighbors of the fake images.

On Test I, pathologist 1 and 2 were able to find only 2 fake out of the 25 sets, and 3 out of the 25, respectively. This is indicative of the images' quality because we argue that pathologists should be less challenged to find fake images amongst real ones, since having other references to compare with facilitates the task. Figure 7 shows Test II in terms of false positive vs true positive, and we can see that pathologist classification is close to random.

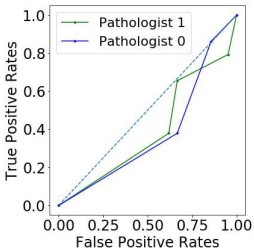

Figure 7: ROC curve of Pathologists' classification for images in Test II. The near random classification performance from both expert pathologists suggests that generated tissue images do not present artifacts that give away the tissue as generated.

The pathologists mentioned that the usual procedure is to work with larger images with bigger resolution, but that the generated fake images were of a quality, that at the $224 \times 224$ size used in this work, they were not able to differentiate between real and fake tissue.

## 5. Conclusion

We presented a new approach to the use of machine learning in digital pathology, using GANs to learn cancer tissue representations. We assessed the quality of the generated images through the FID metric, using the convolutional features of a Inception-V1 network and quantitative cellular information of the tissue, both showed consistent state-of-the-art values of 16.65 and 9.86. We showed that our model allows high level interpretation of its latent space, even performing linear operations that translate into feature tissue transformations. Finally, we demonstrate that the quality of the generated images do not allow pathologists to reliably find differences between real and generated images.

As future work, this model could be extended to achieve higher resolutions of 1024x1024 as TMAs, further experimentation with pathologists and non-experts classification on generated tissue, as well as studies of borderline cases (e.g. atypia) between generated images and pathologists interpretations.

We are most interested in extending our model so it can provide novel understanding of the complex tumor microenvironment recorded in the WSIs, and this is where we think the tissue representation learning properties of our model is key. Being able to encode real tissue patches from WSIs and introducing an encoder into the model that allows us to project real tissue onto the GAN's latent space.

### Acknowledgments

We would like to thank Joanne Edwards and Elizabeth Mallon for helpful insights and discussions on this work.

We will also like to acknowledge funding support from University of Glasgow on A.C.Q scholarship, K.Y from EPSRC grant EP/R018634/1., and R.M-S. from EPSRC grants EP/T00097X/1 and EP/R018634/1.

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

## Appendix A. Code

We provide the code at this location.

## Appendix B. Hinge vs Relativistic Average Discriminator

In this section we show corresponding generated images and loss function plots for Relativistic Average Discriminator model and Hinge Loss model.

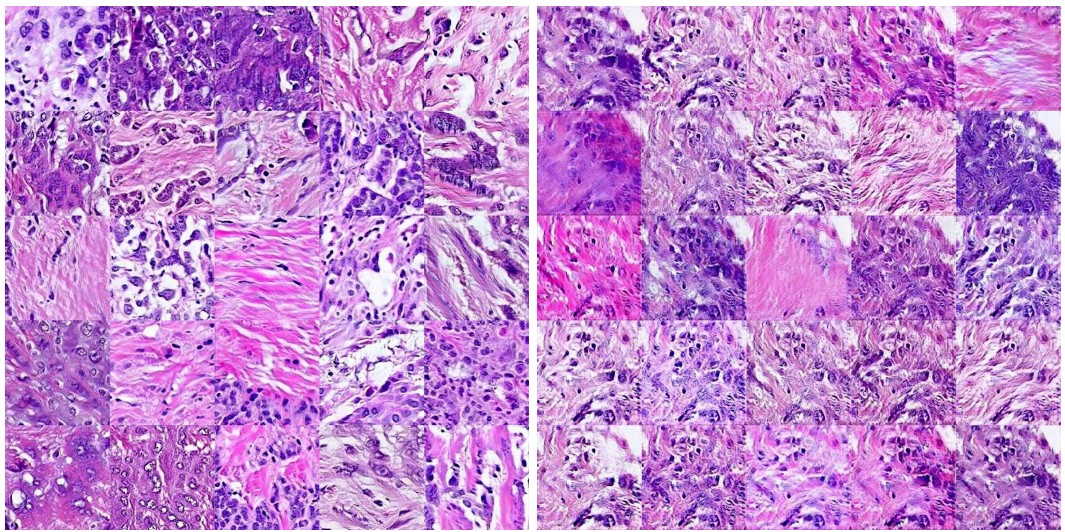

Figure 8: Left grid images correspond to Relativistic Average Discriminator model vs right grid images from the Hinge loss model. We can see that the Relativistic Average model is able to reproduce cancer tissue characteristics compared to Hinge loss, which does not.

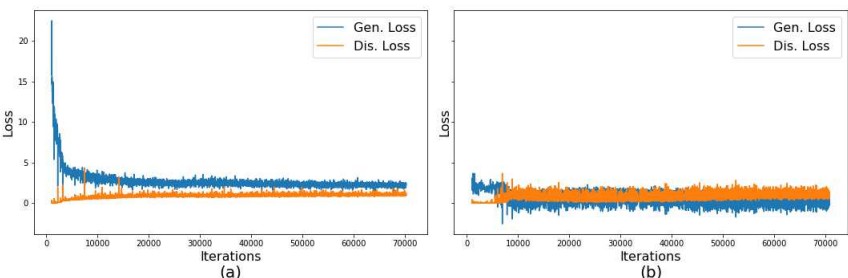

Figure 9: (a) Generator and Discriminator loss functions of the Relativistic Average Discriminator model, (b) Generator and Discriminator loss functions from the Hinge loss model. Here we capture the corresponding loss functions to the images in Figure 8, both of them converge but only Relativistic Average Discriminator produces meaningful images.

## Appendix C. Mapping Network and Style Mixing Regularization Comparison

To measure the impact of introducing a mapping network and using style mixing regularization during training, we provide different figures of the latent space $w$ for two PathologyGANs, one using these features and another one without them.

Figures 10, 11, and 12 capture the clear difference in the latent space ordering with respect to the counts of cancer cells in the tissue image. Without a mapping network and style mixing regularization the latent space $w$ shows a random placement of the vectors subject to the tissue images, when these two elements are introduced moving the selected vector int the latent space from quadrant $IV$ to quadrant $II$ results into increasing the number of cancer cells in the tissue.

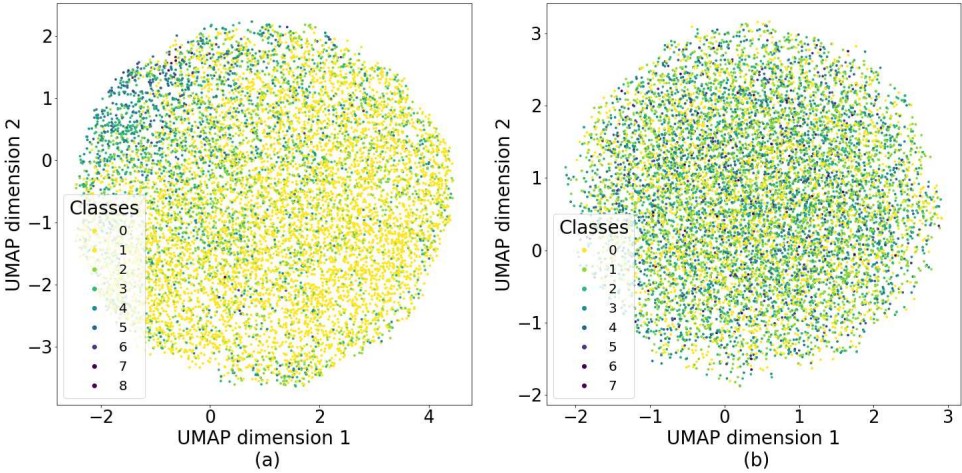

Figure 10: Latent space $w \in \mathbb{R}^{200}$ visualization of $10,000$ vector samples on a UMAP reduced space of 2 dimensions, each vector's generated image was label using CRImage and annotated with the respective class subject to the count of cancer cells in the image. Class 0 accounts for images with the lowest count cancer cells, on the other extreme Class 8 accounts for images with the larger count. (a) corresponds to latent space of a PathologyGAN with a mapping network and style mixing regularization, and (b) to a PathologyGAN without these two features. We show that in model (b) vector samples are randomly placed in the latent space $w$, where in (a) vector samples increasingly concentrate in the quadrant $II$ as we increase the number of cancer cells in the tissue images.

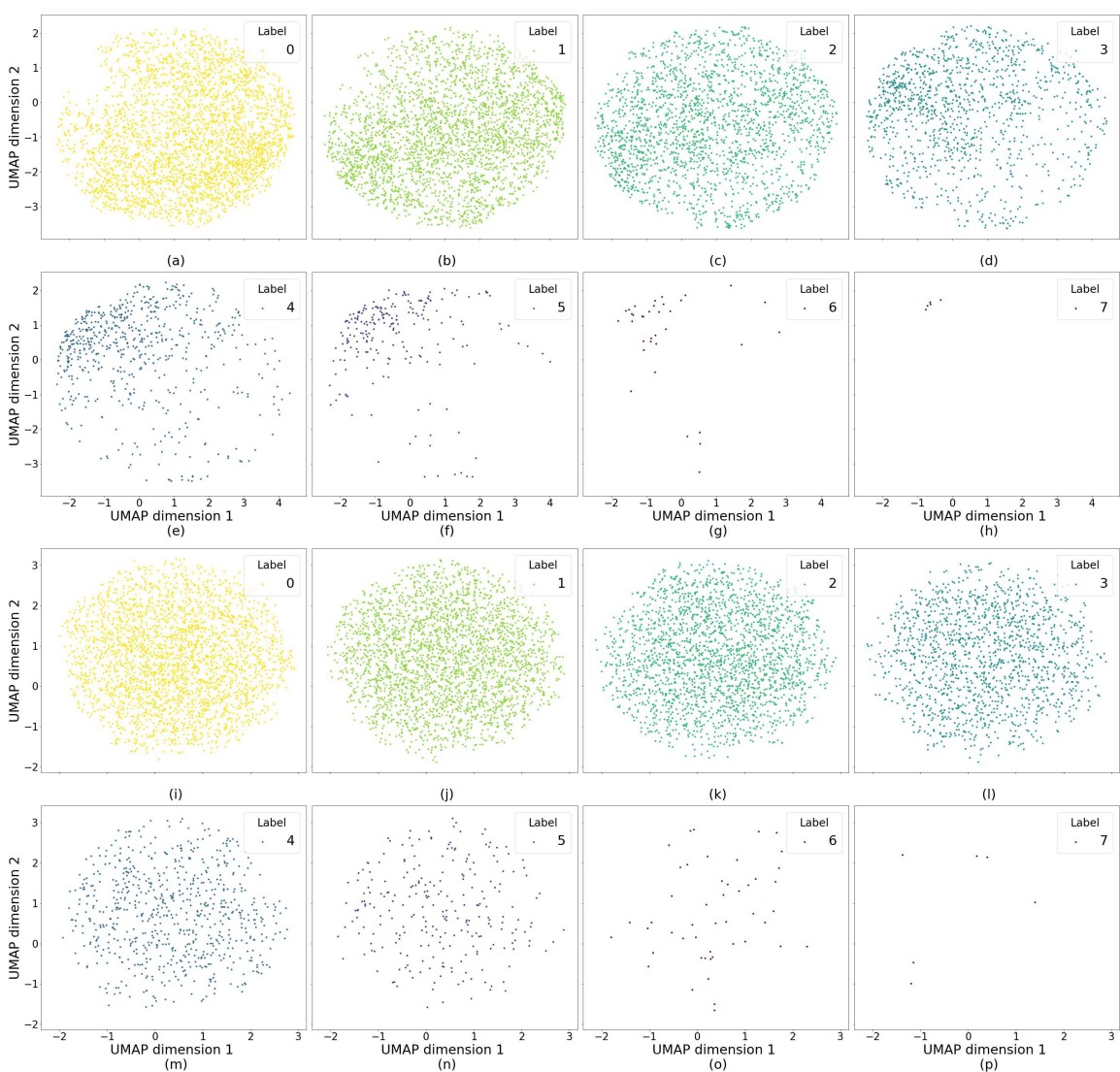

Figure 11: Comparison of the latent space $w$ for two different PathologyGAN models, (a-h) include a mapping network and style mixing regularization, and (i-p) do not include them. Each sub-figure shows datapoints only related to one of the classes, and each class is subject to the count of cancer cells in the tissue image, (a) and (i) [class 0] are associated to images with the lowest number of cancer cells, (h) and (p) [class 8] with the largest. In the model (a-h) images with increasing number of cancer cells correspond to proportionally moving to quadrant $II$ in the 2 dimensional space , where (i-p) are randomly placed.

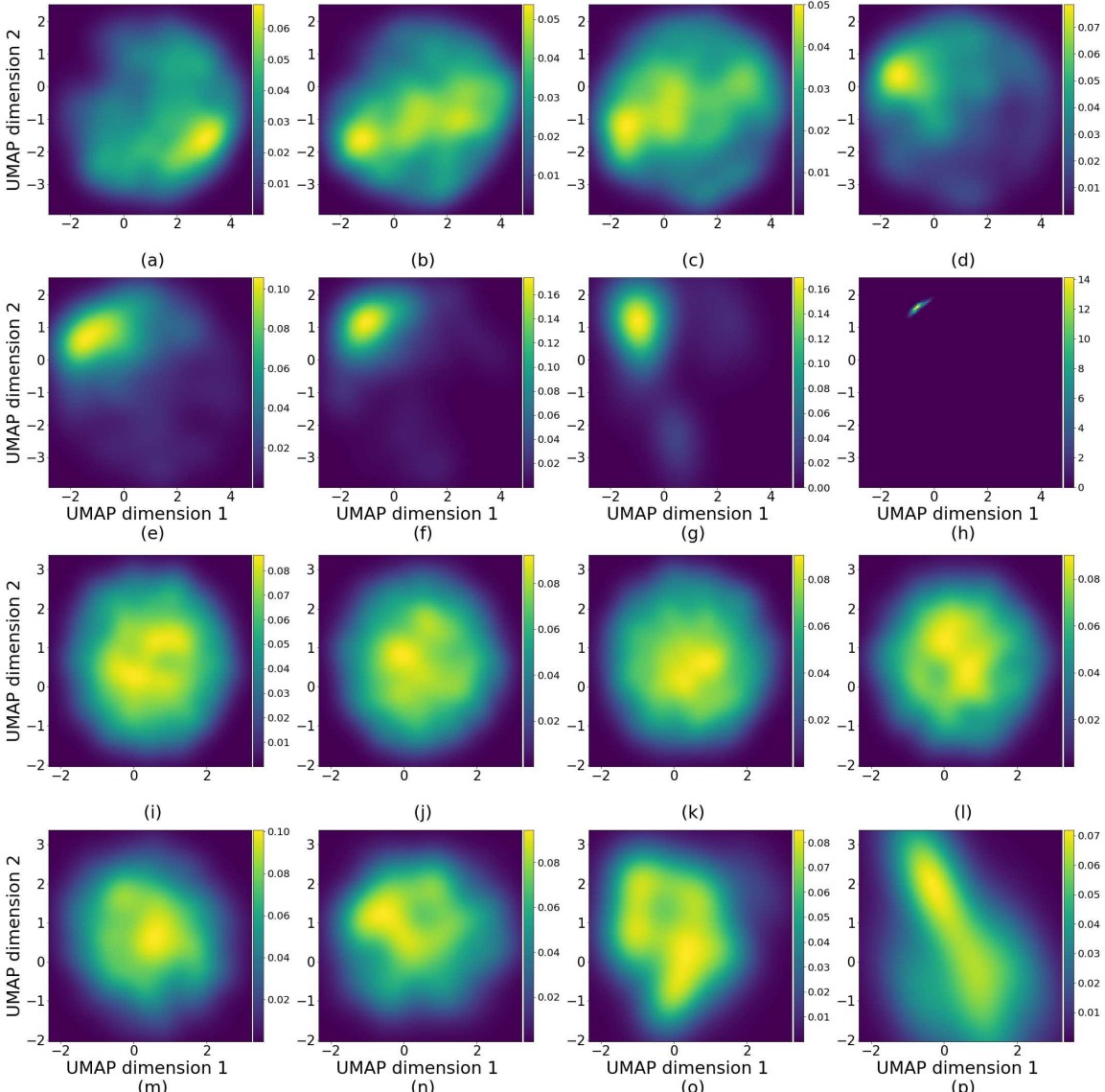

Figure 12: Comparison of the latent space $w$ for two different PathologyGAN models, (a-h) include a mapping network and style mixing regularization, and (i-p) do not include them. Each sub-figure shows the density of datapoints only related to one of the classes, and each class is subject to the count of cancer cells in the tissue image, (a) and (i) [class 0] are associated to images with the lowest number of cancer cells, (h) and (p) [class] with the largest. In the model (a-h) images with increasing number of cancer cells correspond to proportionally moving to quadrant $II$ in the 2 dimensional space , where (i-p) are randomly placed.

## Appendix D. Vector Operation Samples

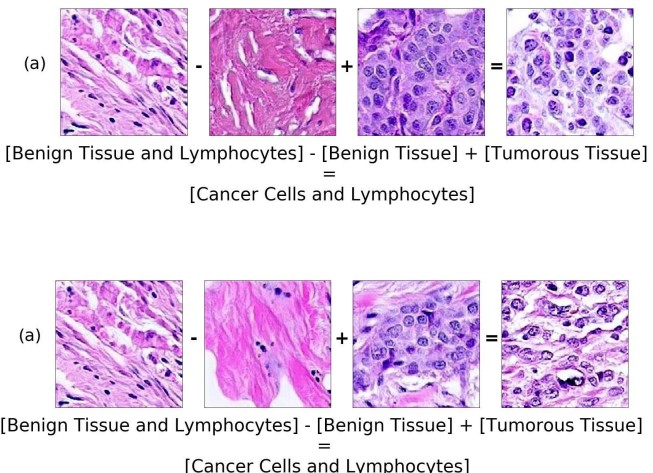

Figure 13: Samples of vector operations with different images, all operations correspond to: Benign tissue and lymphocytes- benign tissue + tumorous tissue = cancer cells and lymphocytes.

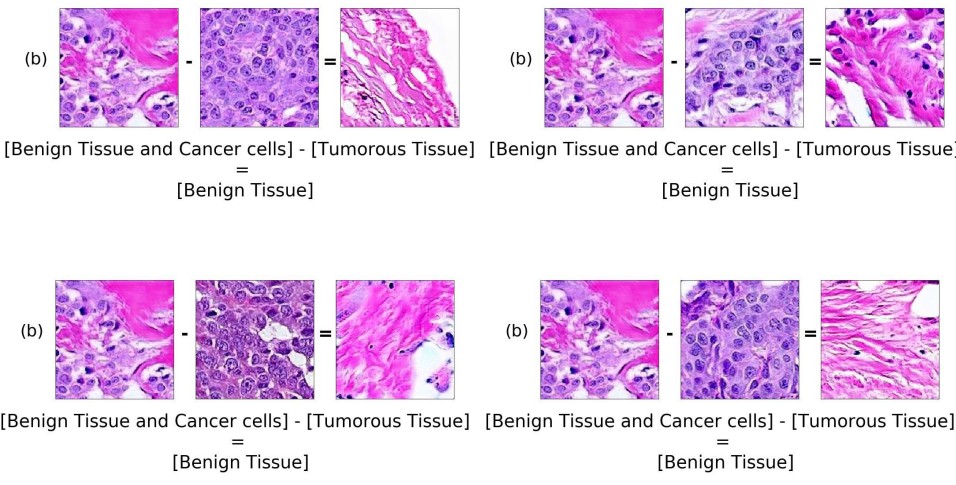

Figure 14: Samples of vector operations with different images, all operations correspond to: Benign tissue with patches of cancer cells - tumorous = benign tissue.

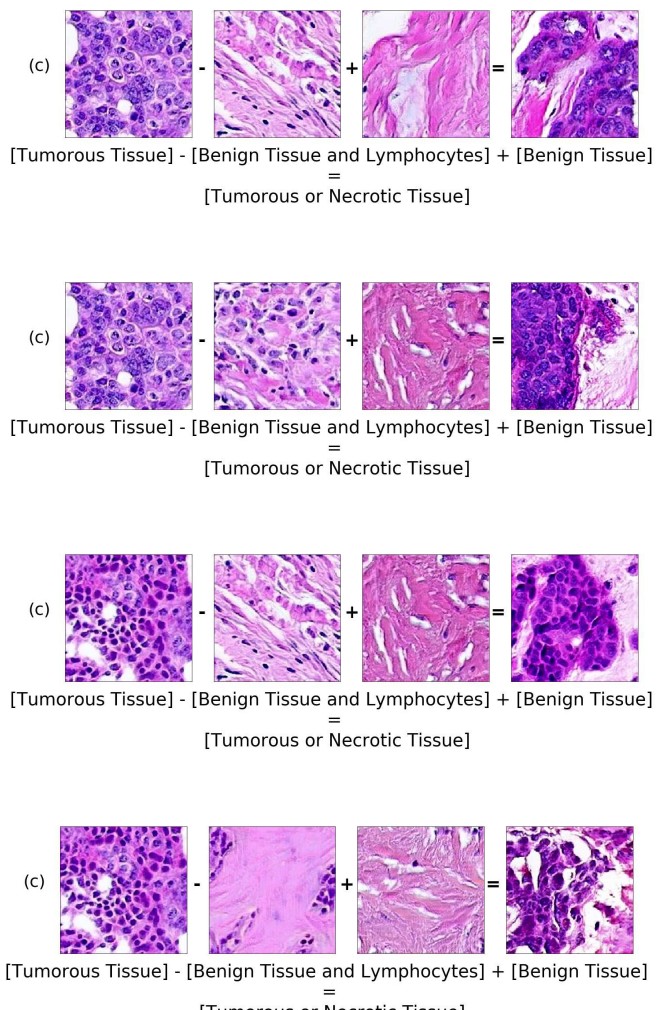

Figure 15: Samples of vector operations with different images, all operations correspond to: Tumorous tissue with lymphocytes - benign tissue with lymphocytes + benign tissue = tumorous or necrotic tissue.

## Appendix E. PathologyGAN at 448x448

We include in this section experimental results of a $448 \times 448$ image resolution model. We trained this model for 90 epochs over approximately five days, using four NVIDIA Titan RTX 24 GB.

Over one model the results of Inception FID and CRImage FID were 29.53 and 203 respectively. We found that CRImage FID is highly sensitive to changes in the images since it looks for morphological shapes of cancer cells, lymphocytes, and stroma in the tissue, at this resolution the generated tissue images don't hold the same high quality as in the

$224 \times 224$ case. As we capture in the **Conclusion** section, this is an opportunity to improve the detail in the generated image at high resolutions.

Figure 16 show three examples of comparisons between (a) PathologyGAN images and (b) real images. Additionally, the representation learning properties are still preserved in the latent space. Figure 17 captures the density of cancer cells in the $448 \times 448$ tissue images as previously presented for the $224 \times 224$ case in **Appendix C.**

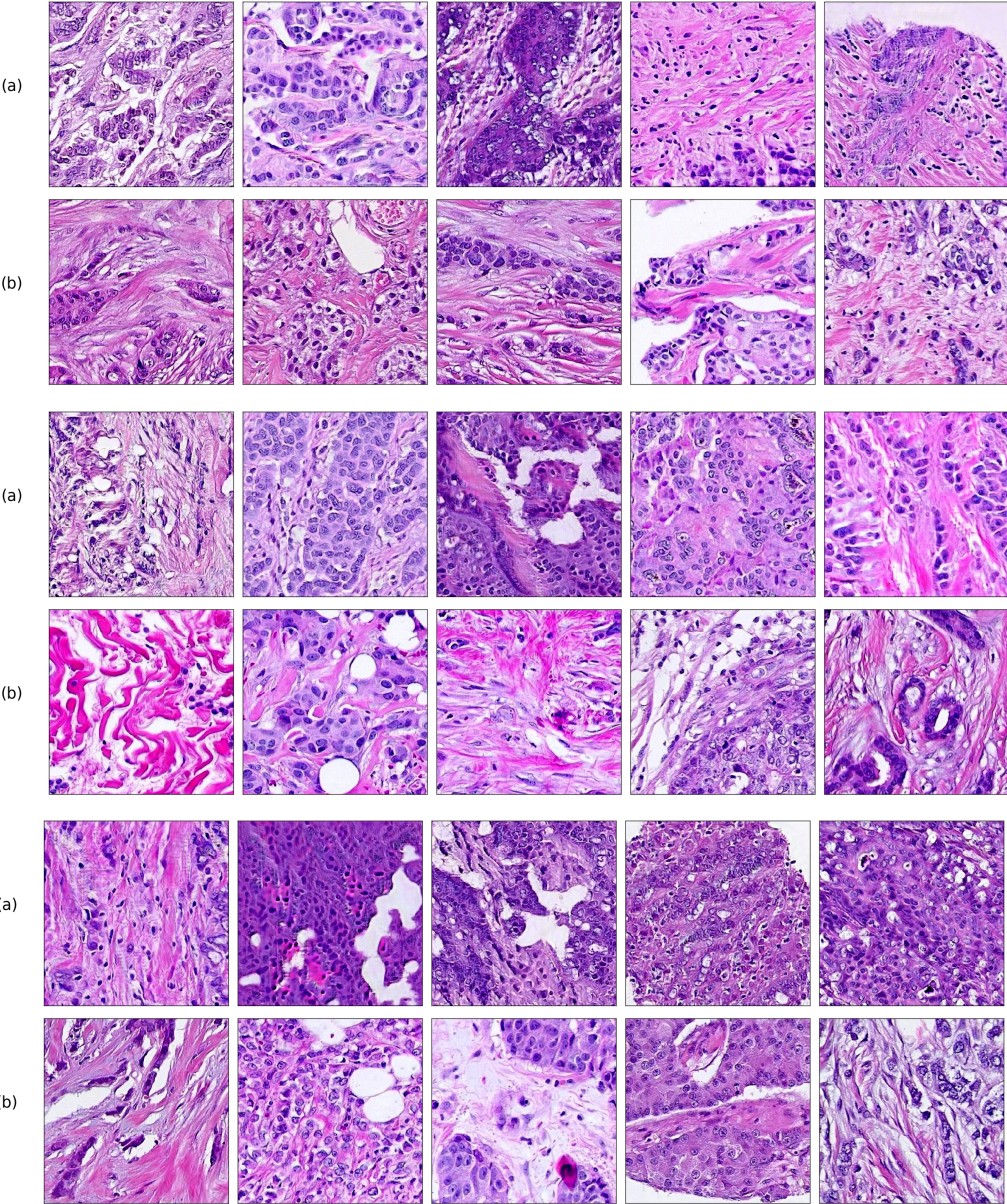

Figure 16: (a): Images ($448 \times 448$) from PathologyGAN trained on H&E breast cancer tissue. (b): Real images.

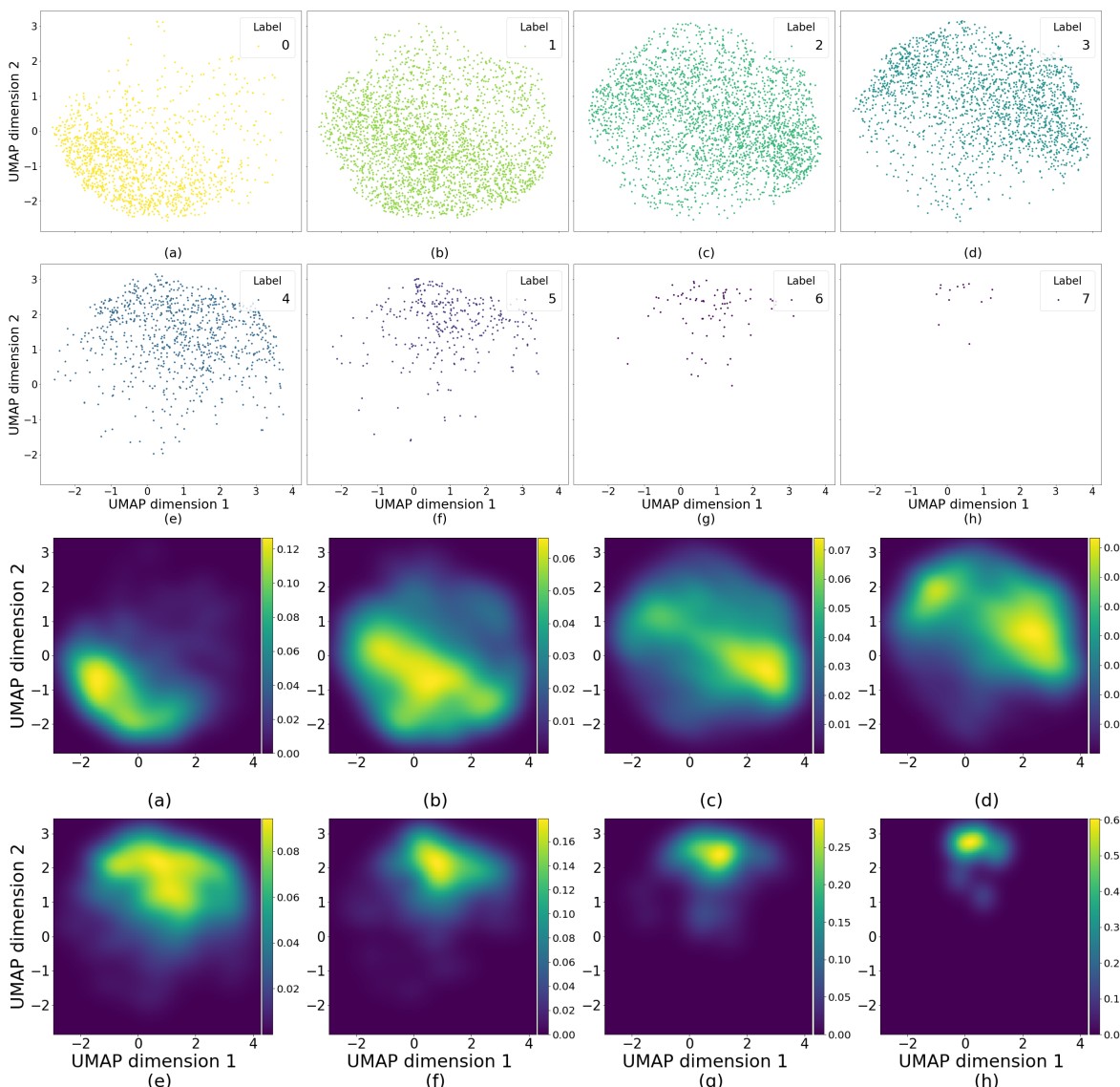

Figure 17: Scatter and density plots of $448 \times 448$ samples on the UMAP reduced representation of the latent space $w$. Each subfigure (a-h) belongs to samples of only one class, where each class represents a range of counts of cancer cells in the tissue image. (a) accounts for images with the lowest number of cancer cells and (h) corresponds to images with the largest count, subfigures from (a) to (h) belong to increasing number of cancer cells.

## Appendix F. GAN evaluation metrics for digital pathology

In this section, we investigate how relevant GAN evaluation metrics perform on distinguishing differences in cancer tissue distributions. We center our attention on metrics that are model agnostic and work with a set of generated images. We focus on Fréchet Inception distance (FID), Kernel Inception Distance (KID), and 1-Nearest Neighbor classifier (1-NN) as common metrics to evaluate GANs. We do not include Inception Score and Mode Score because they do not compare to real data directly, they require a classification network on survival times and estrogen-receptor (ER), and they have also showed lower performance when evaluating GANs (Barratt and Sharma, 2018; Xu et al., 2018).

(Xu et al., 2018) reported that the choice of feature space is critical for evaluation metrics, so we follow these results by using the 'pool_3' layer from an ImageNet trained Inception-V1 as a convolutional feature space.

We set up two experiments to test how the evaluation metrics capture:

- Artificial contamination from different staining markers and cancer types.

- Consistency when two sample distributions of the same database are compared.

### F.1. Detecting changes in markers and cancer tissue features

We used multiple cancer types and markers to account for alterations of color and shapes in the tissue. Markers highlight parts of the tissue with different colors, and cancer types have distinct tissues structures. Examples of these changes are displayed in Figure 18.

We constructed one reference image set with 5000 H&E breast cancer images from our data sets of NKI and VGH, and compared it against another set of 5000 H&E breast cancer images contaminated with other markers and cancer types. We used three types of marker-cancer combinations for contamination, all from the Stanford TMA Database (Marinelli et al., 2008): H&E - Bladder cancer, Cathepsin-L - Breast cancer, and CD137 - Lymph/Colon/Liver cancer.

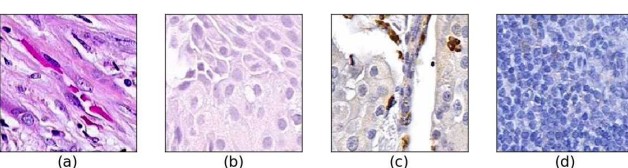

Figure 18: Different cancer types and markers. (a) H&E Breast cancer, (b) H&E Bladder cancer, (c) Capthepsin-L Breast cancer, and (d) CD137 Bone marrow cancer. We can see different coloring per marker, and tissue architecture per cancer type.

Each set of images was constructed by randomly sampling from the respective marker-cancer type data set, which is done to minimize the overlap between the clean and contaminated sets.

Figure 19 shows how (a) FID, (b) KID, (c) 1-NN behave when the reference H&E breast cancer set is measured against multiple percentage of contaminated H&E breast cancer sets. Marker types have a large impact due to color change and all metrics capture this except for

1-NN. Cathepsin-L highlights parts of the tissue with brown colors and CD137 has similar color to necrotic tissue on H&E breast cancer, but still far from the characteristic pink color of H&E. Accordingly, H&E-Bladder has a better score in all metrics due to the color stain, again expect for 1-NN. Cancer tissue type differences are captured by all the metrics, which shows a marker predominance, but we can see that on the H&E marker the differences between breast and bladder types are still captured.

In this experiment, we find that FID and KID have a gradual response distinguishing between markers and cancer tissue types, however 1-NN is not able to give a measure that clearly defines these changes.

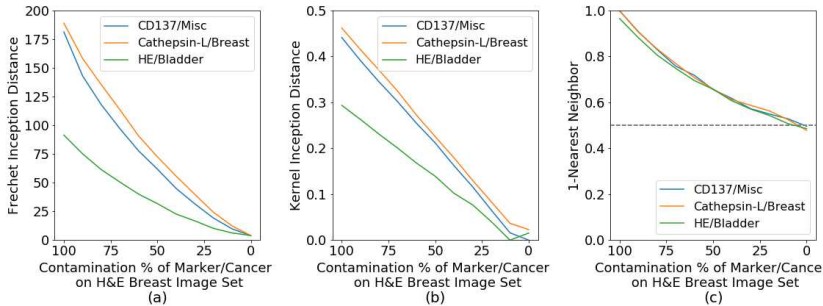

Figure 19: Distinguishing a set H&E Breast cancer images against different contamination of markers and cancer types. For a metric to be optimnal, the value should decreas along with the contamination. (a) corresponds to FID, (b) KID, (c) 1-NN. FID and KID gradually define changes in marker and tissue type, 1-NN does not provide a clear measure of the changes.

### F.2. Reliability on evaluation metrics

Another evaluation we performed was to study which metrics are consistent when two independent sample distributions with the same contamination percentage are compared. To construct this test, for each contamination percentage, we create two independent sample sets of 5000 images and compare them against each other. Again, we constructed these image sets by randomly selecting images for each of the marker-cancer databases. We do this to ensure there are no overlapping images between the distributions.

In Figure 20 we show that (a) FID has a stable performance, compared to (b) KID, and especially (c) 1-NN. The metrics should show a close to zero distance for each of the contamination rates since we are comparing two sample-distributions from the same data set. This shows that only FID has a close to zero constant behavior across different data sets when comparing the same tissue image distributions.

Based on these two experiments, we argue that 1-NN does not clearly represent changes in the cancer types and marker, and both KID and 1-NN do not give a constant reliable measure across different markers and cancer types. Therefore we focused on FID as the most promising evaluation metrics.

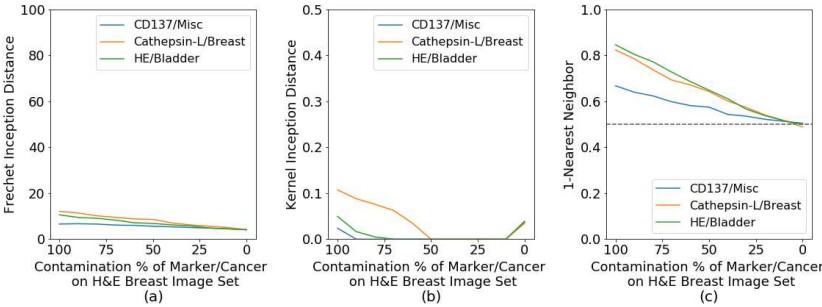

Figure 20: Consistency of metrics when two independent sets of images with the same contamination are compared. Consistent metrics should be close to zero for each of the contamination rates. (a) FID, (b) KID, and (c) 1-NN, we can see that FID is the metric that shows a close to zero constant measure.

## Appendix G. Pathologists Tests

We provide in here examples of the test taken by the pathologists:

- Test I: Sets of 8 images - Pathologists were asked to find the only fake image in each set.

- Test: II: 10 Individual images - Pathologists were asked to rate all individual images from 1 to 5, where 5 meant the image appeared the most real.

Additionally we include the complete tests and solutions.

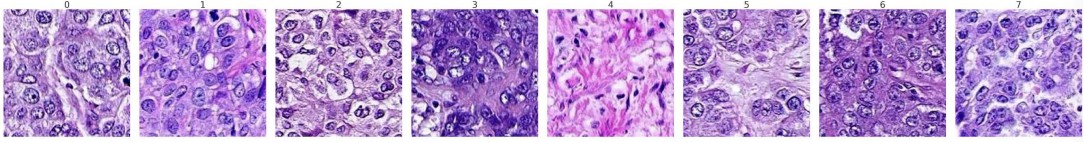

Figure 21: Example of Test I.

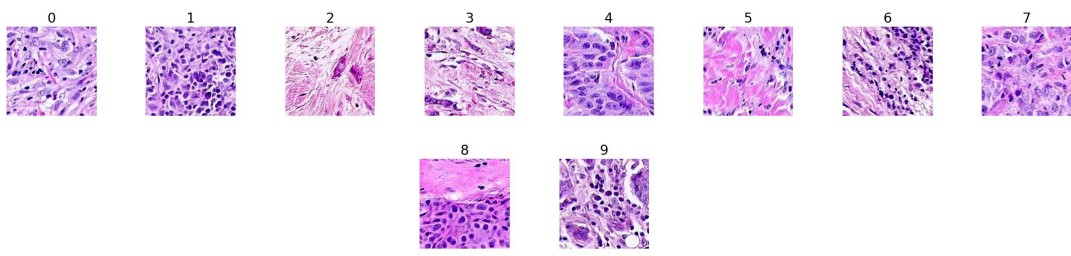

Figure 22: Two examples of Test II.

## Appendix H.  Images

We show here two types of figures:

- Hand-selected fake images with real Inception-V1 closest neighbors.

- Fake images with the smallest distance to real Inception-V1 closest neighbors.

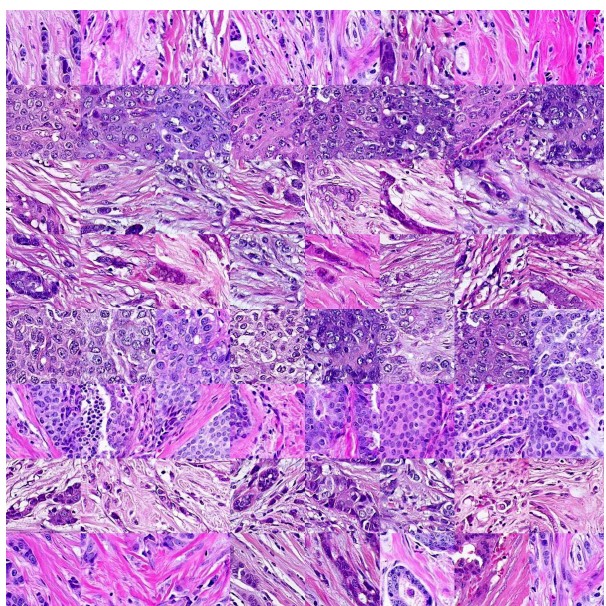

Figure 23:  Hand-selected fake images: for each row, the first image is a generated one, the remaining seven images are close Inception-V1 neighbors of the fake image.

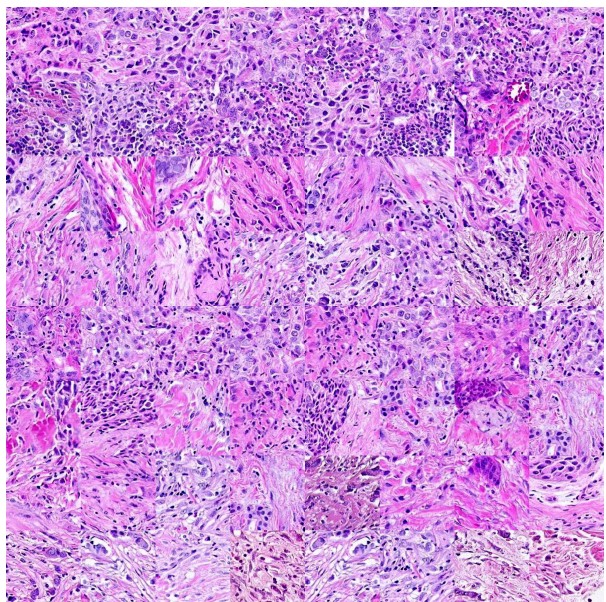

Figure 24: Minimum distance fake images:For each row, the first image is a generated one, the remaining seven images are close Inception-V1 neighbors of the fake image.

## Appendix I. Model Architecture

| Mapping Network $M : z \to w$ |
| :---: |
| $z \in\sim \mathbb{R}^{200} \sim \mathcal{N}(0, I)$ |
| ResNet Dense Layer and ReLU, $200 \to 200$ |
| ResNet Dense Layer and ReLU, $200 \to 200$ |
| ResNet Dense Layer and ReLU, $200 \to 200$ |
| ResNet Dense Layer and ReLU, $200 \to 200$ |
| Dense Layer, $200 \to 200$ |

Table 2: Mapping Network Architecture details of PathologyGAN model.

| Generator Network $G : w \to x$ |
| :---: |
| Dense Layer, adaptive instance normalization (AdaIN), and leakyReLU
$200 \to 1024$ |
| Dense Layer, AdaIN, and leakyReLU
$1024 \to 12544$ |
| Reshape $7 \times 7 \times 256$ |
| ResNet Conv2D Layer, 3x3, stride 1, pad same, AdaIN, and leakyReLU 0.2
$7 \times 7 \times 256 \to 7 \times 7 \times 256$ |
| ConvTranspose2D Layer, 2x2, stride 2, pad upscale, AdaIN, and leakyReLU 0.2
$7 \times 7 \times 256 \to 14 \times 14 \times 512$ |
| ResNet Conv2D Layer, 3x3, stride 1, pad same, AdaIN, and leakyReLU 0.2
$14 \times 14 \times 512 \to 14 \times 14 \times 512$ |
| ConvTranspose2D Layer, 2x2, stride 2, pad upscale, AdaIN, and leakyReLU 0.2
$14 \times 14 \times 512 \to 28 \times 28 \times 256$ |
| ResNet Conv2D Layer, 3x3, stride 1, pad same, AdaIN, and leakyReLU 0.2
$28 \times 28 \times 256 \to 28 \times 28 \times 256$ |
| Attention Layer at $28 \times 28 \times 256$ |
| ConvTranspose2D Layer, 2x2, stride 2, pad upscale, AdaIN, and leakyReLU 0.2
$28 \times 28 \times 256 \to 56 \times 56 \times 128$ |
| ResNet Conv2D Layer, 3x3, stride 1, pad same, AdaIN, and leakyReLU 0.2
$56 \times 56 \times 128 \to 56 \times 56 \times 128$ |
| ConvTranspose2D Layer, 2x2, stride 2, pad upscale, AdaIN, and leakyReLU 0.2
$56 \times 56 \times 128 \to 112 \times 112 \times 64$ |
| ResNet Conv2D Layer, 3x3, stride 1, pad same, AdaIN, and leakyReLU 0.2
$112 \times 112 \times 64 \to 112 \times 112 \times 64$ |
| ConvTranspose2D Layer, 2x2, stride 2, pad upscale, AdaIN, and leakyReLU 0.2
$112 \times 112 \times 64 \to 224 \times 224 \times 32$ |
| Conv2D Layer, 3x3, stride 1, pad same, $32 \to 3$
$224 \times 224 \times 32 \to 224 \times 224 \times 3$ |
| Sigmoid |

Table 3: Generator Network Architecture details of PathologyGAN model.

| Discriminator Network $C : x \rightarrow d$ |
|:---:|
| $x \in \mathbb{R}^{224 \times 224 \times 3}$ |
| ResNet Conv2D Layer, 3x3, stride 1, pad same, and leakyReLU 0.2
$224 \times 224 \times 3 \rightarrow 224 \times 224 \times 3$ |
| Conv2D Layer, 2x2, stride 2, pad downscale, and leakyReLU 0.2
$224 \times 224 \times 3 \rightarrow 122 \times 122 \times 32$ |
| ResNet Conv2D Layer, 3x3, stride 1, pad same, and leakyReLU 0.2
$122 \times 122 \times 32 \rightarrow 122 \times 122 \times 32$ |
| Conv2D Layer, 2x2, stride 2, pad downscale, and leakyReLU 0.2
$122 \times 122 \times 32 \rightarrow 56 \times 56 \times 64$ |
| ResNet Conv2D Layer, 3x3, stride 1, pad same, and leakyReLU 0.2
$56 \times 56 \times 64 \rightarrow 56 \times 56 \times 64$ |
| Conv2D Layer, 2x2, stride 2, pad downscale, and leakyReLU 0.2
$56 \times 56 \times 64 \rightarrow 28 \times 28 \times 128$ |
| ResNet Conv2D Layer, 3x3, stride 1, pad same, and leakyReLU 0.2
$28 \times 28 \times 128 \rightarrow 28 \times 28 \times 128$ |
| Attention Layer at $28 \times 28 \times 128$ |
| Conv2D Layer, 2x2, stride 2, pad downscale, and leakyReLU 0.2
$28 \times 28 \times 128 \rightarrow 14 \times 14 \times 256$ |
| ResNet Conv2D Layer, 3x3, stride 1, pad same, and leakyReLU 0.2
$14 \times 14 \times 256 \rightarrow 14 \times 14 \times 256$ |
| Conv2D Layer, 2x2, stride 2, pad downscale, and leakyReLU 0.2
$14 \times 14 \times 256 \rightarrow 7 \times 7 \times 512$ |
| Flatten $7 \times 7 \times 512 \rightarrow 25088$ |
| Dense Layer and leakyReLU, $25088 \rightarrow 1024$ |
| Dense Layer and leakyReLU, $1024 \rightarrow 1$ |

Table 4: Discriminator Network Architecture details of PathologyGAN model.

