# OpenReview forum: "PathologyGAN: Learning deep representations of cancer tissue"
_MIDL.io/2020/Conference — MIDL 2020_

### Official Review · AnonReviewer1 · 2020-03-03
**Representation learning and data generation for pathology images.**

**Rating:** 4
**Confidence:** 4
**Recommendation:** Oral, Poster

**Summary:**

Authors presented a GAN based pathology image generation method that combines ideas from BigGAN, StyleGAN, and Relativistic Average Discriminator.

They aim to disentangle the style and content via StyleGAN and increases the generated image quality via RAD.

To illustrate the effect of individual model choices, they conducted rigorous testing and present visually pleasing results. They also illustrate space arithmetic examples that look impressive.




**Strengths:**

The paper is overall very well written and the ideas introduced in the paper is well explained. Authors explained their specific algorithmic decisions via both referring to the related literature and also justify them via experiments. In this sense, I think the study is very rigorous

The visual results look very good. Illustration of latent space arithmetic is very good.

The application is important in medical imaging where their is a scarcity of data. But more importantly, in m opinion, defining such latent arithmetic is important in understanding the behavior of the models and making them interpretable.


**Weaknesses:**

Mixing regularization is not explained in detail. The paragraph after equation (2) and (3) is not as clear as the rest of the text. The authors did not clarify why they picked layers 1 and 6 as the points of input entry to the generator network.  The claim "style mixing regularization encourages the generator to localize the high-level features of images in the latent space" is not explained well and supported by experiments or proper citations. It is not clear, why did the authors use 2 different latent representations (z_1 and z_2) rather than one as in the original StyleGAN paper. Last but not least, Figure 2 does not illustrate the entry of 2 inputs to the generator network, so it is a bit confusing.

The diagnostic impact should also be measured. Did the authors conduct a study, where they generated benign and malignant patches and asked the pathologist for their diagnosis? I understand that pathologists, most of the time, do not make a decision over a limited field of view patch; therefore, this experiment can be biased or hard to conduct. The authors can design a test setup using latent space arithmetic (like figure 6) and ask pathologist reading. It would also be interesting to see where the pathologist changes their decision from benign to malignant in the series of images in Figure 5.


**Detailed Comments:**

Authors should be more clear about how they divided the data into training and testing sets. Did the authors form the set of patches first, and then conduct randomization overall patches; or did they conduct randomization over the patients. They should have conduct patient-wise data partitioning. Please clarify

How much data is required to train such a network? Authors mentioned that they used ~250K patches for training, but did they ever try only to use a smaller subset of these images to train models. How does the data size impact performance? This information can be beneficial for researchers who would like to apply this method to their datasets.

**Justification Of Rating:**

The authors presented a thorough study with rigorous testing and visually pleasing results. They well-justified their claims via experiments.

The application is important in medical imaging where their is a scarcity of data. But more importantly, in m opinion, defining such latent arithmetic is important in understanding the behavior of the models and making them interpretable and predictable.



**Paper Type:**

both

**Special Issue:**

yes

---

> ### Author Response · Authors · 2020-03-26
> **Answer to Reviewer 1**
>
>
> First of all, we would like to thank the reviewers and AC for their effort in providing the reviews given the current situation, thank you. We appreciate the reviewers’ constructive comments and the care in which the reviews were done.
>
> - ‘Authors should be more clear about how they divided the data into training and testing sets. ... They should have conducted patient-wise data partitioning. Please clarify‘:
> We divided the train and test sets over the total tissue patches, not according to patients. Our goal was to verify that we learned tissue representations, using the test set to quantify the model with FID scores. We do not use the patient tissue images for survival outcome prediction or any other study of diversity between patients, and for this reason we argue that doing the split on tissue patches is adequate.
>
> To further clarify, if we were to use representations of real tissue images for patient survival outcome, we agree that the proper process to split train and test sets will be over patients.
>
> - ‘How much data is required to train such a network? ... How does the data size impact performance? This information can be beneficial for researchers who would like to apply this method to their datasets.‘:
> We believe that the amount of data we used to train our model is enough to provide good performance. We haven’t studied carefully the relationship between data size and performance, but in an initial study we were able to train a model with only 108K tissue images from the NKI cohort and the model showed high quality images. Nevertheless,  we decided to move forward with all available data in order to ensure all possible diversity of our representations.
>
> (Xu et al. 2016) provides good reference on this matter, Appendix C of this paper has a study on the impact of number of samples and overfitting. They showed that 2000 samples are enough for a GAN to not overfit and provide good representations.
>
> Regrading Style mixing regularization:
> We would like to clarify some of the reviewer’s questions regarding style mixing regularization and provide further justification on some of the statements.
>
> - ‘The claim "style mixing regularization encourages the generator to localize the high-level features of images in the latent space" is not explained well and supported by experiments or proper citations. It is not clear, why did the authors use 2 different latent representations (z_1 and z_2) rather than one as in the original StyleGAN paper.’:
> We’ll add the reference for the original StyleGAN paper on these sentences. In section 3.1 of the paper, the authors explain style mixing regularization in the first paragraph, using two latent vectors (z_1 and z_2) to further localize the styles.
>
> - ‘The authors did not clarify why they picked layers 1 and 6 as the points of input entry to the generator network.‘:
> During style mixing regularization, one layer of all possible layers in the generator is randomly chosen, all layers before the chosen one use the latent vector w_1, and the chosen layer and its consecutives use the latent vector w_2. We use the sentence (e.g. on a generator of ten layers and being six the randomly selected layer, w1 would feed layers one to six and w2 seven to ten) as an example of this procedure.
>
> - ‘Figure 2 does not illustrate the entry of 2 inputs to the generator network, so it is a bit confusing.‘:
> We agree that Figure 2 could better illustrate the structure of the generator. We will work to improve this figure in the final version.
>
> - ‘The diagnostic impact should also be measured. Did the authors conduct a study, where they generated benign and malignant patches and asked the pathologist for their diagnosis? ... It would also be interesting to see where the pathologist changes their decision from benign to malignant in the series of images in Figure 5.‘:
> This an exciting future work, we thank the review for this great suggestion. We agree that it will be interesting to study how the latent space of our model overlaps with pathologists interpretations.
>
> Giovanni  Ciriello,  Martin  L.  Miller,  B ̈ulent  Arman  Aksoy,  Yasin  Senbabaoglu,  Nikolaus Schultz,  and Chris Sander.  Emerging landscape of oncogenic signatures across human Cancers. Nature Genetics, 45(10):1127–1133, 2013.

---

### Official Review · AnonReviewer4 · 2020-03-11
**An interesting application of GAN method in digital pathology**

**Rating:** 3
**Confidence:** 4
**Recommendation:** Oral

**Summary:**

In the paper, the authors proposed a GAN application that allows extracting features from images and shows/ understand differences between cancer and benign areas. The FID metric was used to evaluate the quality of the generated images. The presented method is interesting. The GAN application in Digital Pathology becomes popular last time.


**Strengths:**

The paper presented an interesting idea of the GAN application. The method is well described and easy to follow. Authors present pathologist evaluation as well they used the FID metric to evaluate the quality of generated patches.

**Weaknesses:**

The paper presented an interesting idea of GAN application however it is not clear how it can be used in the future.

The abstract of the paper is confusing. The authors used data extracted from ~600 TMA. However, in the abstract, they present a number of patches that is useless and can be confused. A number of patches that were used for training can be easily increased by additional augmentation and is not representative for readers. In that place, the author should focus on a number of original data (WSI or TMA).

**Justification Of Rating:**

Presented work is interesting and novel. The main advantage of the paper is two folds evaluation : (a) by FID metric and (b) by pathologists.  The method is well described and easy to follow, however, it is not clear how it can be used in the future.

**Paper Type:**

methodological development

**Questions To Address In The Rebuttal:**

The abstract of the paper is confusing. The authors used data extracted from ~600 TMA. However, in the abstract, they present a number of patches that is useless and can be confused. A number of patches that were used for training can be easily increased by additional augmentation and is not representative for readers. In that place, the author should focus on a number of original data (WSI or TMA).

Minor comment:
- Some citations have double brackets.
- Some equations do not have id numbers.

**Special Issue:**

no

---

> ### Author Response · Authors · 2020-03-26
> **Answer to Reviewer 4**
>
>
> First of all, we would like to thank the reviewers and AC for their effort in providing the reviews given the current situation, thank you.
>
> We appreciate the reviewers’ constructive comments and the care in which the reviews were done. We would like to take advantage of some of the feedback to further explain and clarify the motivations and future applications of our work.
>
> The motivation for our work is to develop methods that could lead to a better understanding of phenotype diversity between/within tumors. We hypothesize that this diversity could be quite substantial given the highly diverse genomic and transcriptomic landscapes observed in large scale molecular profiling of tumors across multiple cancer types (Cirello et al. 2013). We argue that representation learning with GAN-based models is the most promising to achieve our goal for the two following reasons: 1) by being able to generate high fidelity images, a GAN could learn the most relevant descriptions of tissue phenotype. 2) the continuous latent representation learned by GAN could help us quantify differences in tissue architectures free from supervised information.
>
> Therefore, the aim of our paper is to provide a generative model framework for unsupervised representation learning on cancer tissue, a  GAN that identifies tissue features, such as color, texture, cancer cells, lymphocytes, and stroma cells (Figure 4, 5, and 6) and shows a structured and interpretable latent space based on those same features (Figures 4 and 5). The quality of tissue generated from representations are quantified with FID scores. We further verified that there aren't notable artifacts missed by FID scores by reviewing our generated images with expert pathologists.
>
> Moving forward, we are working on developing a method that can encode real images using the latent representations constructed by PathologyGAN. This will allow us to assess phenotypic diversity across multiple cancer types, and link phenotypic diversity in tissue to molecular characterizations and patient outcome.
>
> - ‘The authors used data extracted from ~600 TMA. However, in the abstract, they present a number of patches that are useless and can be confused.’:
> We reported the number of patches as it reflects the size of our training data. We agree that it will be beneficial to clarify how many original TMAs we used, and we will introduce those changes in the abstract in a final version. In addition, we provide a description of the data augmentation process at the ‘PathologyGAN’ section, last paragraph.
>
> - ‘The paper presented an interesting idea of GAN application however it is not clear how it can be used in the future.’:
> With the opening statement, we wanted to elaborate on this concern from the reviewer.
>
> - ‘Minor comment:
> Some citations have double brackets.
> Some equations do not have id numbers.’
> Thank you for bringing these issues up, we will fix them in the final version.
>
> Giovanni  Ciriello,  Martin  L.  Miller,  B ̈ulent  Arman  Aksoy,  Yasin  Senbabaoglu,  Nikolaus Schultz,  and Chris Sander.  Emerging landscape of oncogenic signatures across human Cancers. Nature Genetics, 45(10):1127–1133, 2013.

---

### Official Review · AnonReviewer3 · 2020-03-14
**Very good idea with missed opportunities**

**Rating:** 3
**Confidence:** 5

**Summary:**

In this paper the authors presented  a method for  using GANs  on digital pathology images to  represent and generate cancer tissue generation. It is a very neat idea of generating tissue images in H&E domain.It is critical to be able to see the transformation of tissue from benign to malignant. However I think authors missed a couple critical opportunities because of the way they designed their experiments and the problems they wanted to solve.


**Strengths:**

-Applying GAN on digital pathology images would help understand the disease progression mechanism at tissue level, which is very critical for this domain
-The proposed method has powerful interpretation of latent  space between benign and malignant
-The auto generated fake images are looking reasonable
-The method used for testing the quality is adequate


**Weaknesses:**

-The critical tasks in pathology are not usually determining between benign and malignant tissues, that is usually an easy task for pathologists. To be able to show the power of GAN in generating latent space for borderline scenarios (e.g., atypia) would be a better approach. Only by representing these borderline cases the critical questions might be answered by pathologists
-The experiments done on pathologists are not adequate. Experimental design is not deterministic. Asking pathologist to find the fake image within 25 images of low resolution images is not fair. The pathologists are used to look tissue under microscope.
-Need more people to do experiments, maybe the subjects can be 5 pathologist and 5 non-pathologists will allow more clear comparison

**Justification Of Rating:**

Although the experiments are not adequate the method would be a powerful tool to understand the disease mechanisms from digital pathology image. The method could be improved further by including borderline cases and more participants.

**Paper Type:**

both

**Questions To Address In The Rebuttal:**

-Please provide more clear path for future studies

**Special Issue:**

yes

---

> ### Author Response · Authors · 2020-03-26
> **Answer to Reviewer 3**
>
>
> First of all, we would like to thank the reviewers and AC for their effort in providing the reviews given the current situation, thank you.
> We appreciate the reviewers’ constructive comments and the care in which the reviews were done. We would like to take advantage of some of the feedback to further explain and clarify the motivations and future applications of our work.
>
> The motivation for our work is to develop methods that could lead to a better understanding of phenotype diversity between/within tumors. We hypothesize that this diversity could be quite substantial given the highly diverse genomic and transcriptomic landscapes observed in large scale molecular profiling of tumors across multiple cancer types (Cirello et al. 2013). We argue that representation learning with GAN-based models is the most promising to achieve our goal for the two following reasons: 1) by being able to generate high fidelity images, a GAN could learn the most relevant descriptions of tissue phenotype. 2) the continuous latent representation learned by GAN could help us quantify differences in tissue architectures free from supervised information.
>
> Therefore, the aim of our paper is to provide a generative model framework for unsupervised representation learning on cancer tissue, a  GAN that identifies tissue features, such as color, texture, cancer cells, lymphocytes, and stroma cells (Figure 4, 5, and 6) and shows a structured and interpretable latent space based on those same features (Figures 4 and 5). The quality of tissue generated from representations are quantified with FID scores. We further verified that there aren't notable artifacts missed by FID scores by reviewing our generated images with expert pathologists.
>
> Moving forward, we are working on developing a method that can encode real images using the latent representations constructed by PathologyGAN. This will allow us to assess phenotypic diversity across multiple cancer types, and link phenotypic diversity in tissue to molecular characterizations and patient outcome.
>
> - ‘Please provide a more clear path for future studies’:
> In our opening statement above we intend to address and elaborate on future applications of our work.
>
> - ‘To be able to show the power of GAN in generating latent space for borderline scenarios (e.g., atypia) ... Only by representing these borderline cases the critical questions might be answered by pathologists’:
> We agree that our model could be applied to the diagnostic scenarios described by the reviewer. We feel this warrants an exciting future direction to generative models in histopathology. The patient cohorts for this paper might not be powerful enough to explore atypical cases here, so for the scope of this paper we first began with creating this generative model.
>
> - ‘The experiments done on pathologists are not adequate. … The pathologists are used to look tissue under microscope’:
> We agree that our patch size is far from a real WSI or TMA, therefore cannot completely mimic normal pathologists workflow. We did include a sentence in the submission to acknowledge it: ‘The pathologists mentioned that the usual procedure is to work with larger images with bigger resolution’. However, we would like to clarify that this is not the goal of experiment ‘Test II’. The goal of this experiment is to check if there are notable artifacts in the generated images that were not detectable via FID scoring and cell type density matching. Such artifacts have been observed in other deep generative models such as StyleGAN (Karras et al. 2019), or BigGAN (Brock et al 2019). Our results suggest this is less likely to be a problem in our model. We asked the pathologist to find one fake image within seven other real images, in the Appendix G/Figure 20 we provide an example of the same setup we showed the pathologists. Our approach in providing real images along with fake images was to give pathologists other references so they could compare the images. We hypothesized that, given other references, the pathologists would have an easier time finding unclear patterns in the tissue that gave away generated images.
>
> - ‘Need more people to do experiments, maybe the subjects can be 5 pathologist and 5 non-pathologists will allow more clear comparison’:
> We agree that the results would be more robust with a larger number of pathologists. We decided to focus on pathologists only because of their expertise in analyzing tissue and therefore they would be more likely to find tissue hallucinations or artifacts in the generated images. The goal of these experiments was to check if there are notable artifacts (not detectable via FID scoring) in the generated images.
>
> Giovanni  Ciriello,  Martin  L.  Miller,  B ̈ulent  Arman  Aksoy,  Yasin  Senbabaoglu,  Nikolaus Schultz,  and Chris Sander.  Emerging landscape of oncogenic signatures across human Cancers. Nature Genetics, 45(10):1127–1133, 2013.

---

> > ### Comment · AnonReviewer3 · 2020-03-29
> > **Authors are aware of concerns**
> >
> > I'm glad to hear the authors are aware of the concerns I raised. Couple critical comments:
> > - It would be really interesting if authors can experimentally generate a much larger fake tissue image and present in the paper as additional figure or in supplementary. I would like to see the performance on that image both in terms of computation time and image quality
> > - In terms of clear future path, I expected to hear they would expand their study; for generating WSIs, or for understanding evolution on borderline cases, or experimenting on more crowded subjects including expert pathologists and non-experts.

---

> > > ### Author Response · Authors · 2020-03-30
> > > **Answer to Reviewer 3**
> > >
> > > Thank you for the further comments, please find our responses below:
> > >
> > > - 'It would be really interesting if authors can experimentally generate a much larger fake tissue image and present it in the paper as additional figure or in supplementary. I would like to see the performance on that image both in terms of computation time and image quality'
> > > We have been experimenting with a 448x448 model, we can definitely provide this study on the final version of the paper.
> > >
> > > - 'In terms of a clear future path, I expected to hear they would expand their study; for generating WSIs, or for understanding evolution on borderline cases, or experimenting on more crowded subjects including expert pathologists and non-experts.'
> > > We argue current generative models are not able to achieve high resolution images such as WSI (e.g. 40000x40000 pixels) because most novel GANs or VAE are far from that  resolution(e.g. 1024x1024). Our approach for the future is not the traditional approach for GANs in just generating images, we also want to find meaningful latent representations of real tissue patches (e.g. 224x224 or 448x448 of a region of interest) from the WSIs.
> > >
> > > We are most interested in extending our model so it can provide novel understanding of the complex tumor microenvironment recorded in the WSIs, and this is where we think the tissue representation learning properties of our model is key. Being able to encode real tissue patches from the WSI is the sticking point, so we are working on introducing an encoder into the model that allows us to project real tissue onto the GAN’s latent space.
> > >
> > > While pushing this model improvement, we are currently working with WSIs from The Cancer Genome Atlas (TCGA). More specifically, we are looking at patients (~10k) with matched WSI and molecular profiles including genome and transcriptome. This data could answer questions regarding cancer evolution (inferred from genomic data) and its consequence in tumor phenotype, i.e. as recorded in WSI.
> > >
> > > This is a significant undertaking which limits our capacity to pursue exciting questions around borderline cases. Also, we feel that to properly study borderline cases, one would need a well annotated collection of premalignant lesions with molecular profiling or/and retrospective follow up to confirm their borderline status. We are not aware of such a data source and therefore do not have specific plans in this direction.  However, we would be happy to look into it should such data become available.
> > >
> > > Finally, we agree that testing with more expert pathologists and non experts can make our results more robust. We will seek to conduct this experiment in the near future (perhaps for the special issue if we are recommended).
> > >
> > > That said, we also think that our model could also be a tool for other researchers on their work, such as borderline cases as highlighted by the reviewer or further studying how the latent space of our model overlaps with pathologists interpretations.
> > >
> > > We will update the conclusion and future work section to include these points, as we agree they might be relevant for the  research community.

---

### Official Review · AnonReviewer2 · 2020-03-15
**Modified BigGAN for learning histopathology images**

**Rating:** 2
**Confidence:** 4

**Summary:**

The paper proposes a modified BigGAN for histopathology images. Many experiments have been conducted with small patches. The paper is not structured well and is difficult to follow. Asking pathologists to find the fake images may not be the best way to evaluate the approach. I am not sure what to understand from the ROC curve.

**Strengths:**

A new BigGAN for an interesting field; many experiments. A relatively large number of patches used for evaluation. Different metrics used for evaluation. Pathologists involved in validation which adds value.

**Weaknesses:**

- the paper seems to be hastily written
- paper not structured well
- figure positions within the text do not help
- experiments not clearly analyzed
- many results in the appendix
- explicit applications not shown/motivated

**Justification Of Rating:**

Good paper but not structured well; experiments may need more analysis
The paper seems to be hastily written which results in bad organization and not fully analyzed results.
Pathologists involved in validation which adds value.

**Paper Type:**

validation/application paper

**Questions To Address In The Rebuttal:**

What does the ROC curve is really showing?

**Special Issue:**

no

---

> ### Author Response · Authors · 2020-03-26
> **Answer to Reviewer 2**
>
> First of all, we would like to thank the reviewers and AC for their effort in providing the reviews given the current situation, thank you.
>
> We appreciate the reviewers’ constructive comments and the care in which the reviews were done. We would like to take advantage of some of the feedback to further explain and clarify the motivations and future applications of our work.
>
> The motivation for our work is to develop methods that could lead to a better understanding of phenotype diversity between/within tumors. We hypothesize that this diversity could be quite substantial given the highly diverse genomic and transcriptomic landscapes observed in large scale molecular profiling of tumors across multiple cancer types (Cirello et al. 2013). We argue that representation learning with GAN-based models is the most promising to achieve our goal for the two following reasons: 1) by being able to generate high fidelity images, a GAN could learn the most relevant descriptions of tissue phenotype. 2) the continuous latent representation learned by GAN could help us quantify differences in tissue architectures free from supervised information.
>
> Therefore, the aim of our paper is to provide a generative model framework for unsupervised representation learning on cancer tissue, a  GAN that identifies tissue features, such as color, texture, cancer cells, lymphocytes, and stroma cells (Figure 4, 5, and 6) and shows a structured and interpretable latent space based on those same features (Figures 4 and 5). The quality of tissue generated from representations are quantified with FID scores. We further verified that there aren't notable artifacts missed by FID scores by reviewing our generated images with expert pathologists.
>
> Moving forward, we are working on developing a method that can encode real images using the latent representations constructed by PathologyGAN. This will allow us to assess phenotypic diversity across multiple cancer types, and link phenotypic diversity in tissue to molecular characterizations and patient outcome.
>
> - ‘Explicit applications not shown/motivated’:
> Our motivations and vision for applications are detailed in the opening statement. We also want to further emphasize that our approach is to provide a generative model with representation learning properties for quality tissue generation in histopathology.
>
> - ‘What is the ROC curve really showing?’:
> Figure 7 shows the classification performance of two expert pathologists when asked to classify 50 individual images between real and fake. The goal of this experiment is to check if there are notable artifacts (not detectable via FID scoring) in the generated images that an expert would detect compared to real images. The near random classification performance from both expert pathologists suggests that such artifacts are unlikely to occur in our model. Together with high FID scores and matching densities of cell types, these evidence support our claim that PathologyGAN could capture most relevant descriptions of tissue architectures. We will take this feedback into consideration and we will improve Figure 7 caption including additional explanation and conclusions.
>
> - ‘Figure positions within the text do not help’:
> We appreciate this feedback, we will rearrange the Table 1 and Figure 7 so they are not within the text.
>
> - ‘Many results in the appendix’:
> We included many results in the appendix to provide as much information as possible about our work, to further justify our decisions and conclusions, and to promote transparency with our research.
>
> - ‘Paper not structured well’, ‘Experiments not clearly analyzed’:
> We regret the author’s feedback on the structure of the paper and experiment analysis. We welcome any specific feedback that points out any confusion and could help to improve the manuscript.
>
> Giovanni  Ciriello,  Martin  L.  Miller,  B ̈ulent  Arman  Aksoy,  Yasin  Senbabaoglu,  Nikolaus Schultz,  and Chris Sander.  Emerging landscape of oncogenic signatures across human Cancers. Nature Genetics, 45(10):1127–1133, 2013.

---

### Meta-Review · Area_Chair1 · 2020-04-02
**MetaReview of Paper150 by AreaChair1**

**Rating:** 3
**Recommendation For Accepted Papers:** Poster

**Metareview:**

Reviewers are in general positive about this paper. While there is some unclarity regarding application of this method, there is agreement that the analysis is thorough, and - adding to reviewers from my point of view- findings like the visualization of the latent space through interpolation and vector arithmetic is a very interesting feature. Although the methodological contribution is limited, the work seems to be carefully designed to prevent hallucination and provide a meaningful low dimensional representation of pathology relevant image features, thus making it worthwhile to be discussed at MIDL.

**Paper Type:**

both

**Special Issue:**

no

---

### Decision · Program_Chairs · 2020-04-11

Accept